# Learning Summary Statistics for Bayesian Inference with Autoencoders

**Carlo Albert**
Swiss Federal Institute of Aquatic
Science and Technology, Switzerland
carlo.albert@eawag.ch

**Simone Ulzega**
Zurich University of Applied
Sciences, Switzerland

**Firat Ozdemir, Fernando Perez-Cruz**
Swiss Data Science Center,
Switzerland

**Antonietta Mira**
Università Svizzera italiana, Switzerland
University of Insubria, Italy

## Abstract

For stochastic models with intractable likelihood functions, approximate Bayesian computation offers a way of approximating the true posterior through repeated comparisons of observations with simulated model outputs in terms of a small set of summary statistics. These statistics need to retain the information that is relevant for constraining the parameters but cancel out the noise. They can thus be seen as thermodynamic state variables, for general stochastic models. For many scientific applications, we need strictly more summary statistics than model parameters to reach a satisfactory approximation of the posterior. Therefore, we propose to use the inner dimension of deep neural network based Autoencoders as summary statistics. To create an incentive for the encoder to encode all the parameter-related information but not the noise, we give the decoder access to explicit or implicit information on the noise that has been used to generate the training data. We validate the approach empirically on two types of stochastic models.

*Keywords* Bayesian inference · Autoencoder · Thermodynamic state variables

## 1 Introduction

Mechanistic models are indispensable tools in many fields of research. For reliable predictions, they need to include the dominant sources of uncertainty in the form of adequate noise terms. The resulting *stochastic models* often depend on few parameters that need to be calibrated to observed data. For a probabilistic interpretation of the predictions, it is convenient to adopt the *Bayesian framework* and express our knowledge (or belief) about those parameters in terms of probability distributions. Within this framework, calibration means getting hold of the whole *posterior distribution* of parameters, expressing the combination of prior knowledge and knowledge gained from the observations, which is typically a hard *inverse problem*.

Standard algorithms for sampling from the posterior (such as the family of *Metropolis algorithms*) require a large number of evaluations of the *likelihood function*, expressing the probability density for given observed data, as a function of the model parameters. For all but trivial stochastic models, likelihood evaluations are typically prohibitively expensive as they require a high-dimensional integration over model realizations, either because the model has a large number of unobserved (latent) variables or because the normalizing partition function of the model is unknown (think, e.g., of an Ising-type model). A recent solution is to approximate the posterior by means of *neural density estimators* [Papamakarios and Murray, 2016, Papamakarios et al., 2019, Greenberg et al., 2019]. However, this requires a parametrization of the space of approximating densities, which may lead to biases that are hard to control.

Here, we take the approach of *Approximate Bayesian Computation* (ABC, e.g., [Tavaré et al., 1997, Marin et al., 2012]) - an approximate *sampling* method that is a promising alternative to variational methods if model-simulations are *fast* and parameters are *few*. ABC avoids evaluating the likelihood function, by simulating a large number of model realizations,

for various parameter sets, and accepting or rejecting those sets depending on whether or not the simulated data agrees with the observed data in terms of a given set of *summary statistics*, and within a given *tolerance*. For ABC to be accurate and efficient, we need summary statistics that, respectively, retain most of the parameter-related information and cancel out most of the noise. The latter requirement entails that summary statistics are well-concentrated, for fixed parameter values, which allows for a fast annealing of the tolerance [Albert et al., 2014].

Obvious candidates for such statistics are *parameter estimators*, such as the maximum likelihood estimator (MLE). Hence, several methods for finding summary statistics are based on *parameter regression* (e.g., [Fearnhead and Prangle, 2012, Jiang et al., 2017, Wiqvist et al., 2019]). Parameter estimators are constraining the *location* of the posterior, but not necessarily its *shape*. If the data consists of a large number of independent sample points, the posterior is typically well concentrated around the true parameter values, in which case parameter estimators are near-sufficient, i.e. they contain nearly all the information that is relevant for constraining the posterior. However, in many scientific applications, we only have few realizations (e.g. from a lab experiment) or even just one (e.g. from a field experiment). And although such data sets may consist of many components (e.g. time series), they may be highly correlated. Therefore, models describing such data might produce rather different realizations, even for a *fixed* set of parameter values, and the corresponding posteriors might look rather different as well. In such situations, additional statistics may be required to reach a satisfactory approximation of the posterior.

In order to capture those additional statistics, we suggest to combine regression with reconstruction, i.e. to use neural networks of *Autoencoder-type*. Recently, Autoencoders have been employed to learn *order parameters* [Wetzel, 2017] or *collective variables* [Bonati et al., 2020], which are quantities capable of discriminating between different kinds of behavior of statistical-mechanics models. However, in order to use Autoencoders for the purpose of ABC, we have to modify them such that they encode only the features that carry information about the parameters, but not the noise. To do so we give the decoder access to explicit or implicit noise information. Thus, it creates an incentive for the encoder to encode parameter-related information only.

What distinguishes our approach from other information-theoretic machine learning algorithms such as the ones recently presented in [Cvitkovic and Koliander, 2019] and [Chen et al., 2021], is the possibility to use *explicit* noise information. This makes our approach also applicable in situations where only noise-, but no parameter-information is available, as could be the case in observed rather than simulated data. As an example, we might want to use it to remove rain-features (rain playing the role of the noise) from hydrological runoff time-series in order to distill runoff-features that stem from the catchments themselves. We will examine this application in future publications.

The software written for this project is available on `https://renkulab.io/gitlab/bistom/enca-inca`.

## 2 Summary Statistics

Consider a generic stochastic model, defined by a conditional probability density, $f(\mathbf{x}|\boldsymbol{\theta})$, where $\boldsymbol{\theta} \in \mathbb{R}^p$ is a (low-dimensional) parameter vector and $\mathbf{x} \in \mathbb{R}^N$ a (high-dimensional) output vector. Furthermore, consider a map, $\mathbf{s} : \mathbb{R}^N \to \mathbb{R}^q$, of *summary statistics*, where $q$ is small (of the order of $p$). We require $\mathbf{s}$ to be *asymptotically sufficient*, meaning that

$$I(\boldsymbol{\Theta}, \mathbf{S}) := I(\boldsymbol{\Theta}, \mathbf{s}(\mathbf{X})) = I(\boldsymbol{\Theta}, \mathbf{X}) + \mathcal{O}(1/N), \tag{1}$$

where $I$ denotes the *mutual information* between dependent random variables. Here, $\boldsymbol{\Theta}$ is the parameter prior with given density $f(\boldsymbol{\theta})$ and, for given $\boldsymbol{\theta}$, $\mathbf{X} \sim f(\mathbf{x}|\boldsymbol{\theta})$. Eq. (1) means that, for large data sets, the summary statistics contain almost as much information about the parameters as the whole data set. Furthermore, for ABC to converge efficiently, we require the summary statistics to cancel the noise contained in model-outputs, and thus to be ever more concentrated around a $p$-dimensional manifold (at least locally) as $N$ grows larger[1]. Thus, invoking the law or large numbers, we require the *asymptotic concentration property*

$$H(\mathbf{S}|\boldsymbol{\Theta}) := - \int f(\mathbf{s}|\boldsymbol{\theta})f(\boldsymbol{\theta}) \ln(f(\mathbf{s}|\boldsymbol{\theta})) d\mathbf{s} d\boldsymbol{\theta} \sim -\ln(N), \tag{2}$$

as $N \to \infty$, which is an asymptotic minimal entropy condition. As can be seen from the identity

$$H(\mathbf{S}|\boldsymbol{\Theta}) = H(\mathbf{S}) - I(\boldsymbol{\Theta}, \mathbf{S}), \tag{3}$$

there is a trade-off when adding more summary statistics. It typically increases the mutual information, but also the entropy $H(\mathbf{S})$. Therefore, we expect an optimal number of statistics, which leads to the most concentrated summary statistics and hence the most efficient convergence of ABC.

---

[1]Here we assume that all parameters are identifiable. Otherwise the dimension of the manifold could be less than $p$.

The asymptotic sufficiency and concentration properties allow us to draw an analogy between summary statistics and *thermodynamic state variables*, as already pointed out by [Mandelbrot, 1962]. In statistical mechanics, the concentration property leads to the *equivalence of ensembles* in the thermodynamic limit. Extending this analogy a bit further, we define the *free energy*

$$F_{\boldsymbol{\theta}}(\mathbf{s}) := -\ln \int f(\mathbf{x}|\boldsymbol{\theta}) d\Omega_{\mathbf{s}}(\mathbf{x}), \tag{4}$$

where $\Omega_{\mathbf{s}}(\mathbf{x})$ is the surface measure on the shell $\mathbf{s}(\mathbf{x}) = \mathbf{s}$. For members of the exponential family, the free energy splits into an energy and an entropy term as

$$F_{\boldsymbol{\theta}}(\mathbf{s}) = U_{\boldsymbol{\theta}}(\mathbf{s}) - S(\mathbf{s}). \tag{5}$$

If the data $\mathbf{y}$ is comprised of a large number of *independent* sample points, the free energy will be dominated by the energy term. Hence, the summary statistics will concentrate around a $p$-dimensional submanifold of minimal energy configurations. Since minimum energy is synonymous with maximum likelihood, we can parametrize this manifold with *maximum likelihood estimators* (MLE). Hence, we can encode nearly all information relevant for constraining the parameters with $q = p$ summary statistics. This is no longer the case if $\mathbf{x}$ is *correlated*, in which case the entropy can substantially alter the free energy landscape. If (2) is satisfied, the summary statistics will eventually concentrate locally around a $p$-dimensional submanifold. However, due to correlations, certain features of the output might de-correlate very slowly and lead to broad valleys or multiple modes in the free energy landscape. Multiple modes can even persist in the limit as $N \to \infty$, in which case the model exhibits different *phases*. For such models, $\mathbf{S}$ does not concentrate around a single $p$-dimensional submanifold, and we might need $q > p$ summary statistics, for a good posterior approximation.

## 3 Modified Autoencoders

We want to design Autoencoders implementing both the asymptotic sufficiency criterion (1) as well as the concentration property (2). The *encoder*, $\mathbf{s}(\mathbf{x})$, is supposed to learn the summary statistics. The first $p$ of the $q$ summary statistics are regularized to be parameter regressors, whereby the concentration property is implemented. The auxiliary $q - p$ summary statistics are meant to capture additional information required for constraining the posterior. The *decoder* is fed with the vector $\mathbf{s}$ and, in addition, with either explicit or implicit information on the noise that went into generating the training data, and is supposed to either reconstruct the model outputs or the parameter values. In this manner, the encoder is incentivized to encode all the parameter-related information (sufficiency), but not the noise (concentration). The first architecture we propose (Fig. 1) has a decoder that attempts to reconstruct the model output $\mathbf{x}$. For this architecture, the stochastic model needs to be set up as a deterministic function, $\mathbf{x} = M(\boldsymbol{\theta}, \boldsymbol{\epsilon})$, where the bare noise $\boldsymbol{\epsilon}$ is sampled from a $\boldsymbol{\theta}$-independent distribution. Such a formulation is possible, for *any* stochastic model. However, it is not unique, and the performance of the Autoencoder might depend on the choice. The decoder is then given the *explicit* noise realizations $\boldsymbol{\epsilon}$ that went into the generation of the training data and attempts to reconstruct the model output, i.e. it learns a function $\hat{\mathbf{x}}(\mathbf{s}, \boldsymbol{\epsilon})$. We refer to this architecture as the Explicit Noise Conditional Autoencoder (ENCA). The decoder will not only have to learn the model equations but implicitly also the reconstruction of the true parameter values $\boldsymbol{\theta}$ from $\mathbf{s}$ and $\boldsymbol{\epsilon}$ (to the extent possible). For the loss function, we use the weighted sum of squares

$$\mathcal{L} = \frac{1}{p} \sum_{\alpha=1}^{p} \left( \frac{s_\alpha - \theta_\alpha}{\theta_\alpha} \right)^2 + \frac{1}{N} \sum_{i=1}^{N} \left( \frac{\hat{x}_i - x_i}{x_i} \right)^2. \tag{6}$$

The second architecture (Fig. 2) does not require a separation of bare noise, and is therefore more broadly applicable. It provides *implicit* information on the noise to the decoder in the form of replica of summary statistics encoded from different realizations of the model for *fixed* parameter values. That is, for a given set of parameter values $\boldsymbol{\theta}$, we generate $n$ realizations, $\mathbf{x}^{(j)}$, for $j = 1, \dots, n$, from $f(\mathbf{x}|\boldsymbol{\theta})$, which are then passed individually onto the encoder to predict $n$ summary statistics $\mathbf{s}^{(j)} = \mathbf{s}(\mathbf{x}^{(j)})$. Since we do not have access to the bare noise, we do not attempt to reconstruct the model output $\mathbf{x}$, as such a reconstruction would generally be very poor. Instead, the decoder attempts to aggregate the information in the replica of summary statistics and reconstruct the true parameters, $\hat{\boldsymbol{\theta}}(\{\mathbf{s}^{(j)}\})$. We refer to this architecture as the Implicit Noise Conditional Autoencoder (INCA). If we again regularize the first $p$ components of $\mathbf{s}$ to be parameter regressors, we can set

$$\hat{\theta}_\alpha(\{\mathbf{s}^{(j)}\}) = \frac{\sum_{j=1}^{n} w(\mathbf{s}^{(j)}) s_\alpha^{(j)}}{\sum_{j=1}^{n} w(\mathbf{s}^{(j)})}, \quad \alpha = 1, \dots, p, \tag{7}$$

where $w(\cdot)$ is a nonlinear function to be learned by the decoder based on the auxiliary summary statistics $\{s_\beta^{(j)}\}$, for $\beta = p+1, \dots, q$. If the parameter posterior strongly depends on the particular realization $\mathbf{x}^{(j)}$, for a fixed set of

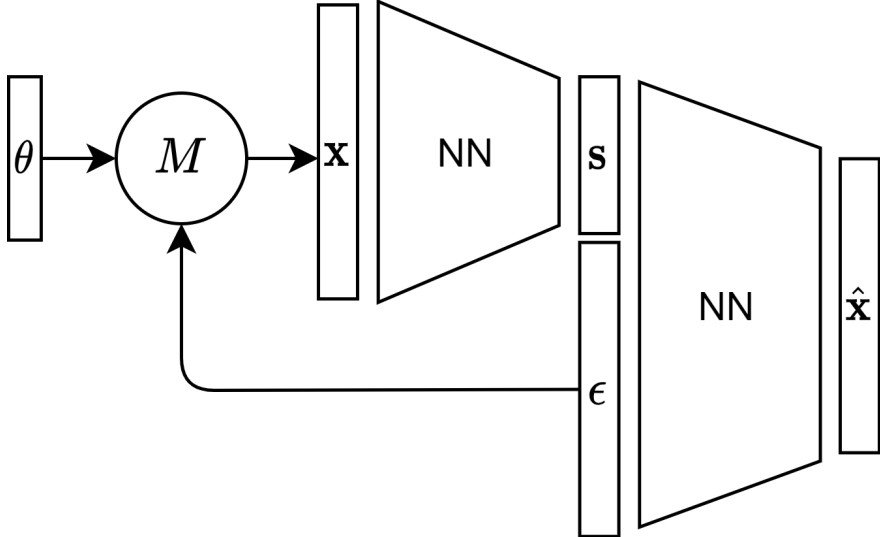

Figure 1: Basic ENCA architecture: The stochastic model needs to be available in the form of a deterministic function, $M$, of a parameter vector $\boldsymbol{\theta}$ and a (bare) noise realization $\boldsymbol{\epsilon}$. The encoder is trained on model realizations $\mathbf{x}$, and produces summary statistics $\mathbf{s}(\mathbf{x})$. The decoder is trained to reproduce the model realizations $\hat{\mathbf{x}}(\mathbf{s}, \boldsymbol{\epsilon})$, based on $\mathbf{s}$ and the same noise realizations that have been used by $M$. Layers within encoder and decoder neural network (NN) blocks can be optimized based on the prior knowledge about $M$.

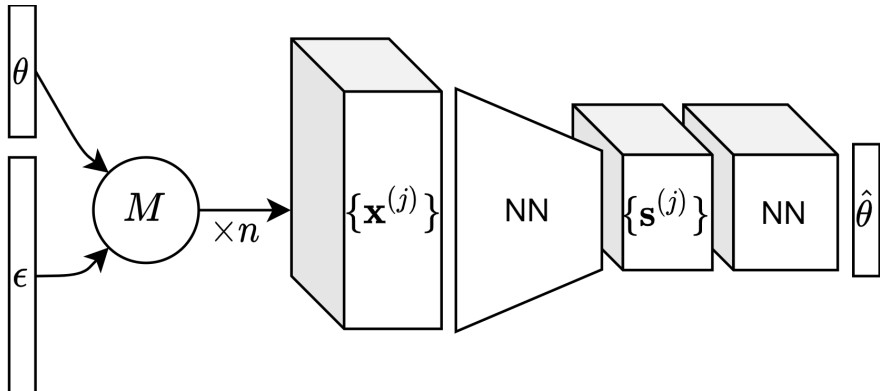

Figure 2: Basic INCA architecture: The encoder is trained on sets of model realizations $\{\mathbf{x}^{(j)}\}$, for fixed parameter values, and produces the corresponding sets of summary statistics $\{\mathbf{s}^{(j)} = \mathbf{s}(\mathbf{x}^{(j)})\}$. The decoder is a NN acting on auxiliary summary statistics $\{s_\beta^{(j)}\}$, for $\beta = p+1, \ldots, q$ to find a weighting function $w$ to reconstruct model parameters, based on eq. (7).

parameters $\boldsymbol{\theta}$, the reliability of the associated parameter regressors $s_\alpha^{(j)}$, for $\alpha = 1, \ldots, p$, will also strongly depend on the particular realization $j$, which should be accounted for with the weights. This is precisely how the decoder incentivizes the encoder to use the auxiliary summary statistics to encode the extra features that distinguish these different realizations. As the loss function we use

$$\mathcal{L} = \sum_{j,\alpha} \left( \frac{s_\alpha^{(j)} - \theta_\alpha}{\theta_\alpha} \right)^2 + \sum_\alpha \left( \frac{\hat{\theta}_\alpha - \theta_\alpha}{\theta_\alpha} \right)^2 . \tag{8}$$

## 4    Results

We demonstrate our method with two types of stochastic iterative map models, for which we have access to the true posterior. The first one is given by the equation

$$x_{n+1} = \alpha f(x_n) + \sigma \epsilon_n , \quad \epsilon_n \sim \mathcal{N}(0,1) \quad \text{i.i.d.}, \quad n = 0, \ldots, N-1 \, (N = 200) , \tag{9}$$

where $f(x)$ is a nonlinear function admitting two stable solutions, e.g. $f(x) = x^2(1-x)$. The deterministic map has a stable fixed point at $x = 0$. For sufficiently large $\alpha$, a second stable fixed point emerges, which, upon further increasing $\alpha$, undergoes a series of period doubling bifurcations eventually leading to chaos (see Appendix 6.1, for more details on the model). Depending on the initial condition $x_0$, the deterministic map will go to either one of the two attractors. This deterministic solution is the *energetically* most favored realization as it corresponds to the zero-noise realization. For sufficiently large noise, entropic effects become more important and a bi-modal free energy surface can occur. Fig. 5 shows the two types of realizations the model exhibits, even for fixed parameter values. As a member of the exponential family, the minimal number of sufficient statistics for this model is bounded. However, the parametrization not being the natural one, we need *three* summary statistics to reach sufficiency, albeit the model only has two parameters, $\boldsymbol{\theta} = (\alpha, \sigma)^T$. A convenient parametrization, for sufficient statistics, is given by eqs.

$$\hat{\alpha}(\mathbf{x}) = \frac{\sum_{n=1}^{N} x_n f(x_{n-1})}{\sum_{n=1}^{N} (f(x_{n-1}))^2} , \tag{10}$$

$$\hat{\sigma}(\mathbf{x}) = \frac{1}{N} \sum_{n=1}^{N} (x_n - \hat{\alpha}(\mathbf{x}) f(x_{n-1}))^2 , \tag{11}$$

$$o(\mathbf{x}) = \frac{1}{N} \sum_{n=1}^{N} (f(x_{n-1}))^2 . \tag{12}$$

The first two statistics are MLEs for the two parameters, inferring $\alpha$ from the auto-correlation and $\sigma$ from the residuals, respectively. The third one, $o(\mathbf{x})$, can be seen as an *order parameter* that is needed to tell the two attractors apart. We have chosen the initial point $x_0$ in between the two attractors, and the prior such that the switching time between them is much longer than the observation time. Thus, there are parameter values $\boldsymbol{\theta}$, for which $F_{\boldsymbol{\theta}}(\mathbf{s})$ is bi-modal, for $\mathbf{s} = (\hat{\alpha}, \hat{\sigma}, o)^T$ (Fig. 4). For these values, the third statistic is not approximated by a function of the other two. Hence it contains information about the parameters that is not contained in the other two statistics. Fig. 3 shows that apart from prior boundary effects, both architectures yield parameter regressors (first two components of $\mathbf{s}$) that are very similar to MLEs (first two components of the sufficient summary statistics (10), (11)). In order to accurately reconstruct the output, the decoder of both ENCA and INCA forces the encoder to also learn a quantity equivalent to the third statistic, although INCA separates the two "phases" less clearly (Fig. 4). Fig. 6 compares ABC-posteriors, for different sets of summary statistics, against a posterior-sample generated with a Metropolis algorithm. Using the encoder-generated summary statistics from the two architectures confirms that both are near-sufficient, albeit the approximate posterior achieved with INCA is a bit less accurate; i.e., wider. Both approximate posteriors are much closer to the true posterior compared to the approximate posterior we get when we only use the two MLE regressors (10) and (11). Hence, our Autoencoder-generated summary statistics will outperform any statistics generated by machine-learning models that are solely based on parameter regression. We have also trained ENCA with only two summary statistics. The encoder then encodes the information about which attractor has been chosen into these two statistics, at the price of a deteriorated parameter regression (Fig. 5). The ABC-posterior resulting from these two statistics is thus closer to the truth than the MLE-posterior, but farther away than the one generated with three ENCA-encoded statistics (Fig. 6).

Notice that, for extremely large $N$, typical realizations would switch between the two attractors sufficiently often for there to be just one type of model behavior, and consequently no need for a third statistic. Hence, this model does *not* exhibit phases. However, the example shows that there may be features in the output of stochastic models that de-correlate very slowly as $N$ grows, requiring auxiliary statistics even for large $N$.

The second example is a stochastic non-linear iterative map model with both additive and multiplicative noise:

$$x_{n+1} = \alpha_n f(x_n) + \epsilon_n , \, \alpha_n \sim \mathcal{U}[\alpha, \alpha + \delta] , \, \epsilon_n \sim \mathcal{U}[0, \epsilon] \quad \text{i.i.d.}, n = 0, \ldots, N-1 \, (N = 200) , \tag{13}$$

and three parameters, $\boldsymbol{\theta} = (\alpha, \delta, \epsilon)^T$. It is an example for the common situation where we have more internal-noise degrees of freedom $(\boldsymbol{\alpha}, \boldsymbol{\epsilon})$ than observed output components $(\mathbf{x})$. Integrating out the unobserved degrees of freedom typically takes us outside of the exponential family, as is the case for this model. According to the Pitman-Koopman-Darmois theorem, we would need a set of summary statistics that grows unbounded with the size of the data $(N)$ to achieve strict sufficiency. However, we expect to be able to compress most of the parameter-related information into few summary statistics nonetheless. We have trained both ENCA and INCA with 3 $(q = p)$ as well as 4, 5 and 6

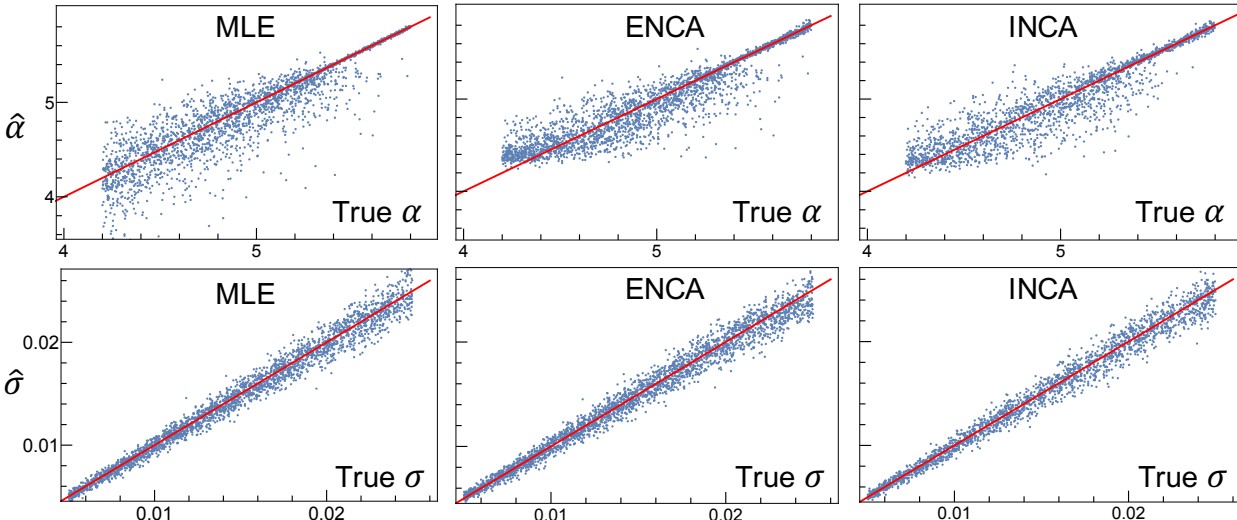

Figure 3: MLEs (using eqs. (10 and (11)) for the parameters of model (9) (left panels), parameter regression from the ENCA- (middle panels), and the INCA-encoder (right panels). Both ENCA and INCA are trained with $q = 3$ summary statistics. The upper row of plots shows a clear accuracy-difference for the two phases of the model.

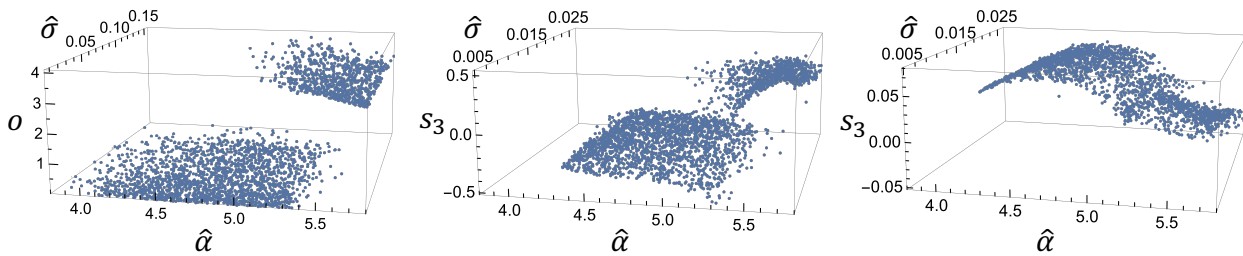

Figure 4: Prior distribution of sufficient summary statistics (10) - (12) (left), and of the latent variables from both ENCA (middle) and INCA (right). The two sheets correspond to the two attractors of the model and overlap on the $(\hat{\alpha}, \hat{\sigma})$-projection (bi-modality).

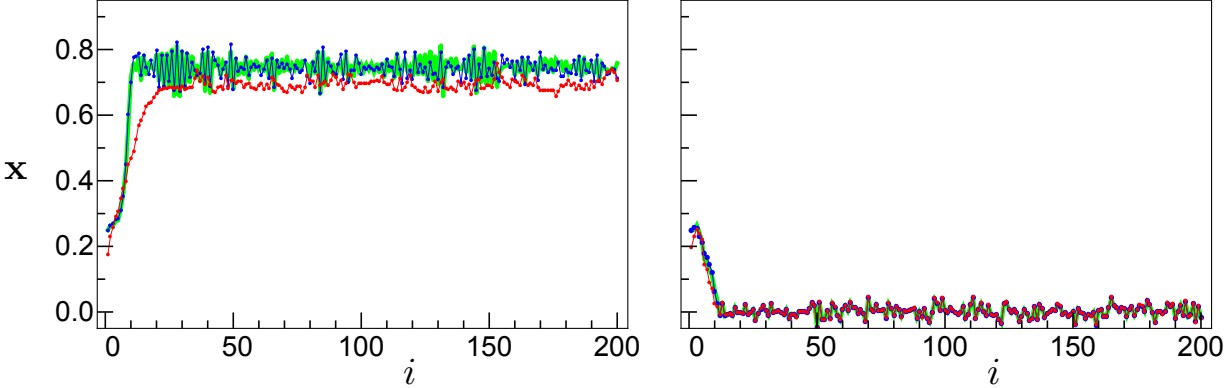

Figure 5: Typical time-series $\mathbf{x}$ generated with model (9) (green) compared against the reconstructions by the ENCA-decoder, using two (red) or three (blue) latent variables. The two time-series were generated with the same set of parameters, $\alpha = 5.3$ and $\sigma = 0.015$. Two statistics are sufficient to encode the information about which attractor has been taken, but at the price of a degraded parameter regression and thus a degraded reconstruction. The data has not previously been used for training the AE.

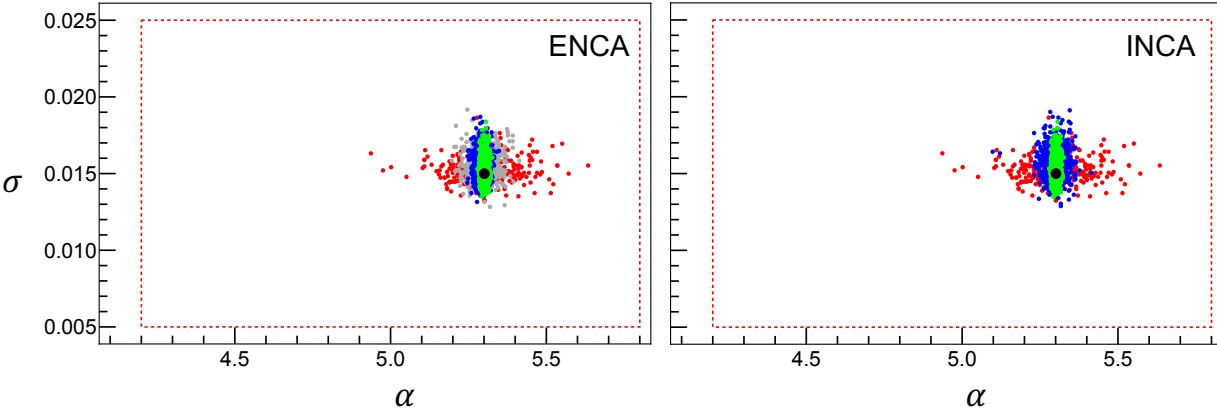

Figure 6: Metropolis-generated posterior for model (9) (green, representing the ground truth) compared against ABC posteriors using all three latent variables (blue) from ENCA (left) and INCA (right) and using only the two MLE-regressors (red). The posterior resulting from only two ENCA-generated summary statistics is shown in grey. The dashed boxes represent the ABC priors, while the black dots represent the true values of the parameters used to generate the synthetic data set.

summary statistics. Fig. 7 shows that, when training ENCA on model (13), adding a fourth statistic has a small yet noticeable effect on the first statistic, i.e. the regressor for $\alpha$. This means that the decoder can facilitate the regression of the encoder, but its main purpose is to create the incentive to encode additional information in the auxiliary statistics. That it does so indeed is shown in Fig. 8. When using only 3 statistics the decoder does not manage the reconstruct the model output well, but with only 4 the reconstruction is almost perfect. Correspondingly, using 4 or 5 statistics leads to approximate posteriors that are more accurate than the one achieved with only 3 statistics. (Fig. 9). Using more than 5 statistics leads to a degradation of the ABC-posterior. Presumably this is because the statistics start to encode more of the noise, which degrades their concentration. Similarly, the approximate posteriors resulting from INCA-generated summary statistics are most accurate when using 5 statistics (Fig. 10).

When deciding about the optimal number of statistics (when the true posterior is not known) the distances to the observations achieved with ABC can be an indication (Fig. 11). According to eq. (3), large distances (lack of concentration) can be the result of either too few statistics (lack of sufficiency) or too many (lack of minimality). The posterior accuracy achieved with ENCA is in line with the achieved ABC distances. Although with INCA the smallest distances are achieved with 3 statistics, the posterior with 5 statistics is marginally better. This is the case because, for dimensional reasons, the lower the number of statistics the smaller the distances tend to be.

## 5 Conclusions

We have given a proof of concept that Autoencoder-like architectures providing additional noise-information to the decoder can be used to learn summary statistics that are both near-sufficient and highly concentrated - two criteria that are required for ABC to work both accurately and efficiently. While the decoder can help the encoder to find good parameter regressors, its main job is to create an incentive for encoding additional information that is relevant for contraining the parameters. We have proposed two types of decoders, one that attempts to reconstruct the outputs and is given explicit noise information, and one that attempts to reconstruct the parameters and is given implicit noise information. In our case studies, the former achieved slightly better results, whereas the latter has a broader range of applicability. For the proof of concept we have chosen members of two types of nonlinear stochastic models. The first model is a member of the exponential family, which allowed us to compare the learned summary statistics against a known set of sufficient statistics. This model also exemplifies that stochastic nonlinear models can exhibit strongly correlated features, which might require us to use more summary statistics than parameters. The second model is an example for the common situation where we have more internal-noise degrees of freedom than observed output components, and lies outside of the exponential family, where no bounded set of sufficient statistics is available. Nevertheless, in our example we show that a very good approximation of the true posterior can be achieved with ABC with a small number of learned summary statistics that is, once again, larger than the number of parameters. As a criterion for the optimal number of summary statistics we suggest to use the distances between simulated and observed summary statistics achieved within ABC.

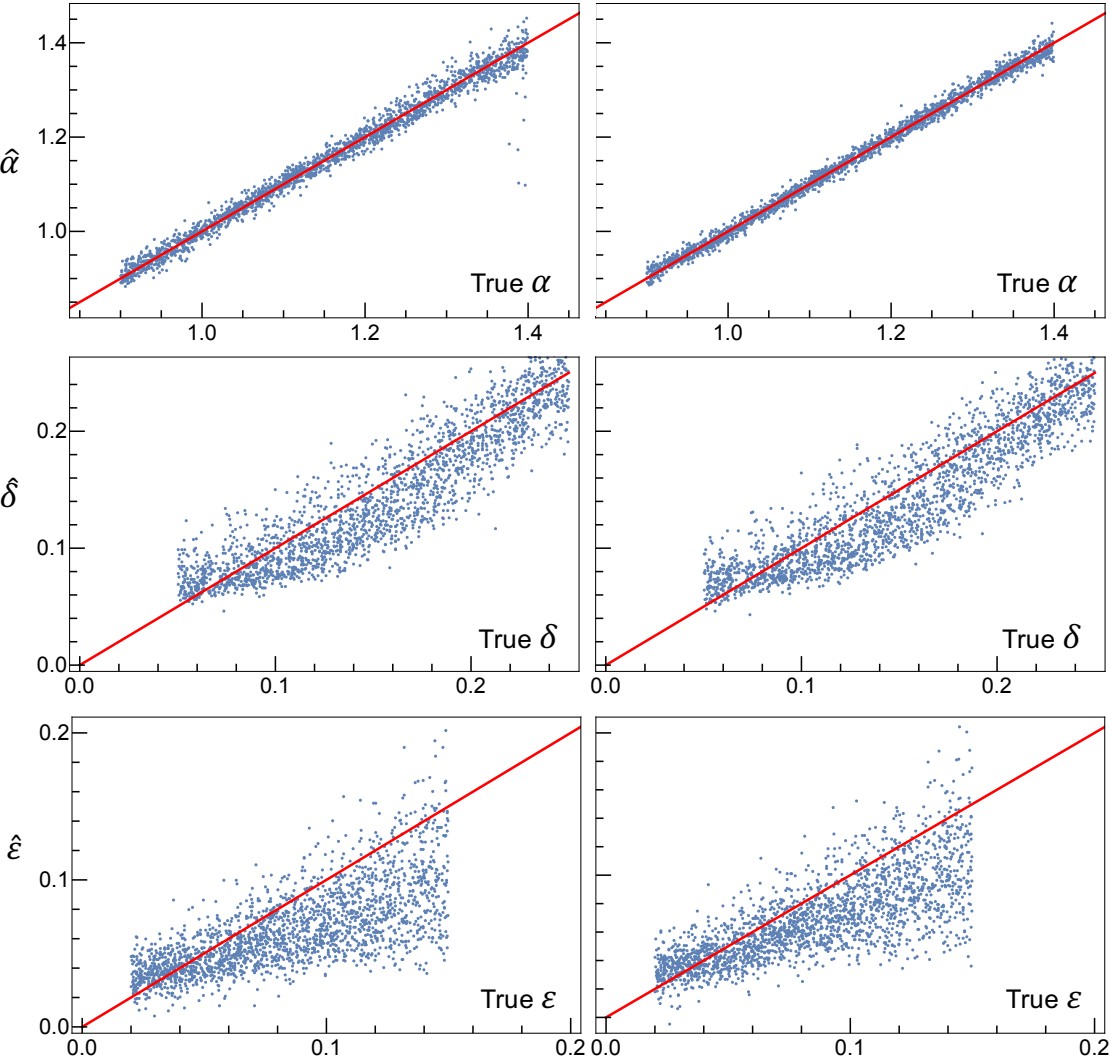

Figure 7: Parameter regression for model (13) from ENCA, using three (left column) or four (right column) latent variables. The contribution of the additional feature to the $\alpha$-marginal is small but noticeable. (Results are similar for INCA.)

Our method is applicable whenever we want to disentangle low-dimensional relevant features from high-dimensional irrelevant (noise) features. Thus, its applicability goes beyond summary statistics learning for ABC inference. An exciting avenue for future research is to apply it to learn relevant low-dimensional features in observed rather than simulated data. Here, we only present a proof of concept. More research is required to explore and validate the applicability of our approach in practical applications.

**Acknowledgements**

This work is part of the BISTOM project, funded by the Swiss Data Science Center.

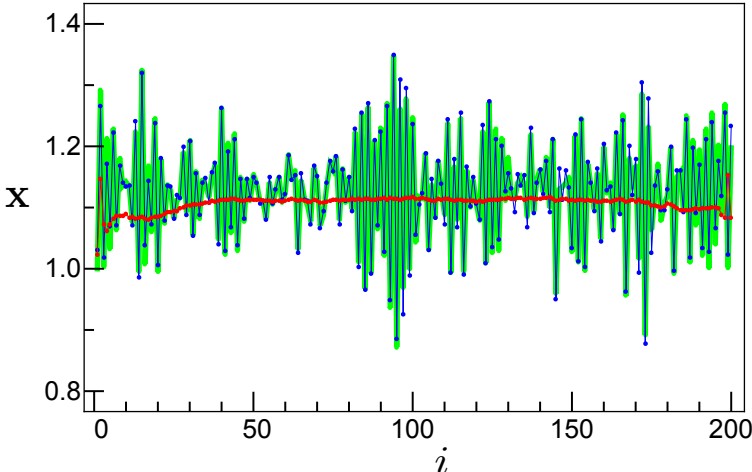

Figure 8: Typical time-series **x** generated by model (13) (green) compared against the reconstruction by the ENCA-decoder, using three (red) or four (blue) latent variables. The contribution of the additional statistic is indisputable. The data has not previously been used for training the AE.

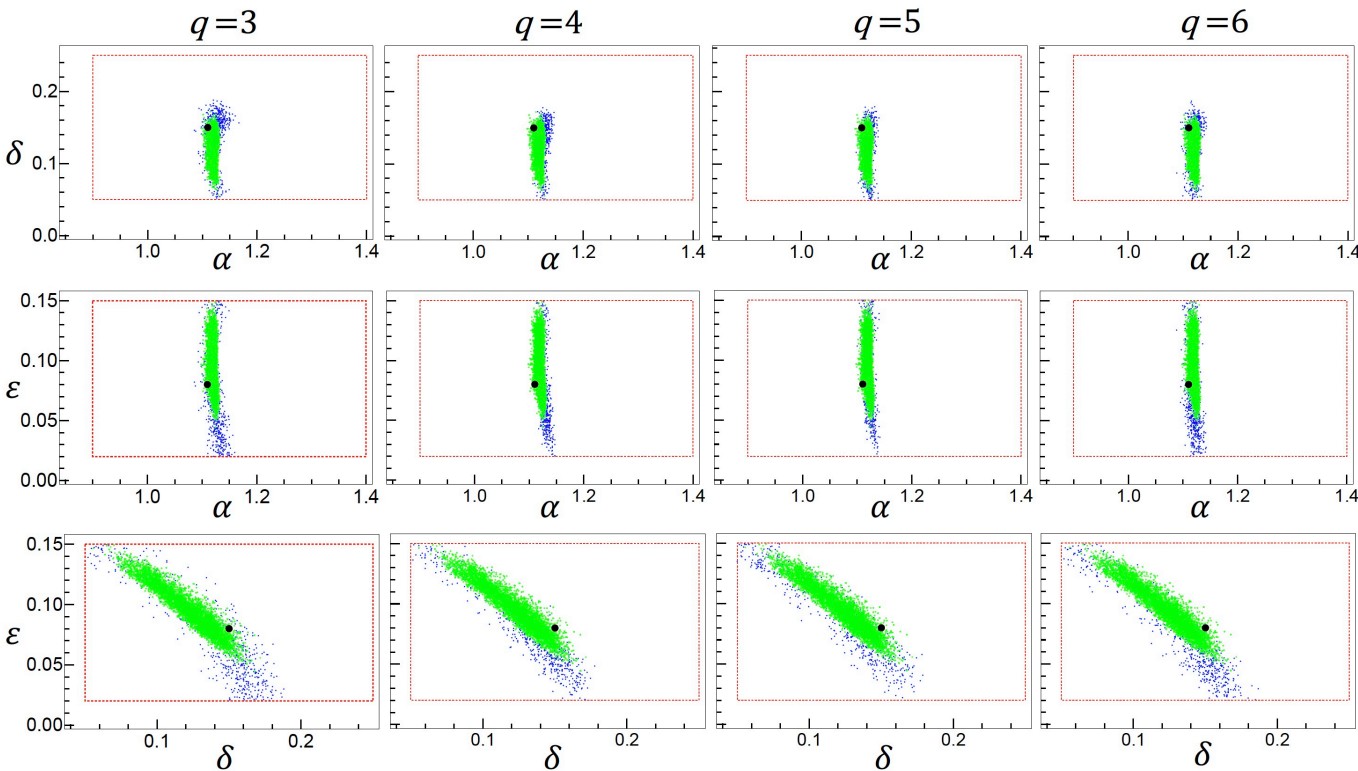

Figure 9: 2D projections of the true posterior for model (13) (green) compared against ABC posteriors (blue) from ENCA using, from left to right columns, three, four, five and six summary statistics ($q = 3, 4, 5, 6$). The dashed boxes represent the ABC priors, while the black dots represent the true values of the parameters used to generate the synthetic data set.

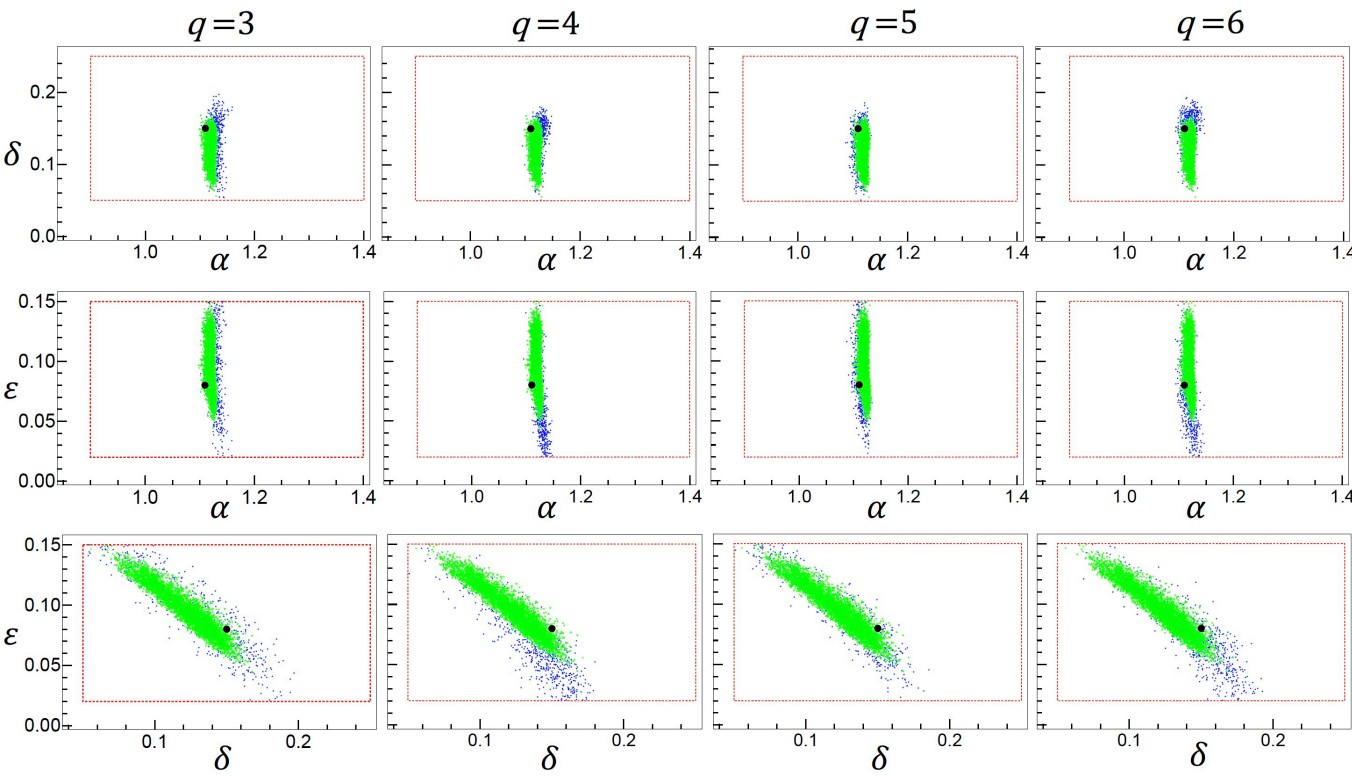

Figure 10: Analogously to Fig.9, the exact posteriors for model (13) (green) are compared against ABC posteriors (blue) from INCA.

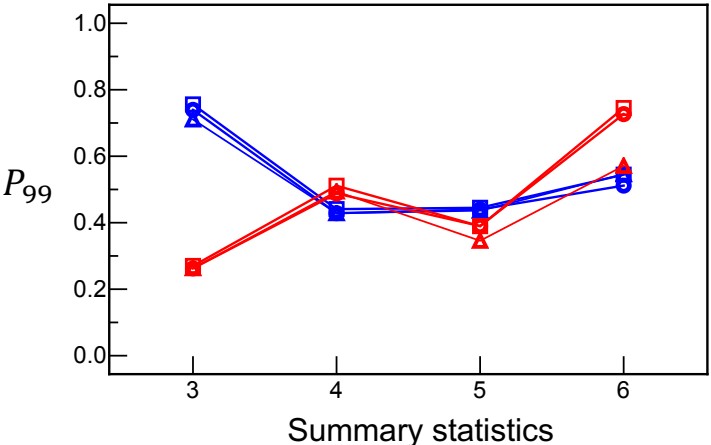

Figure 11: 99%-percentiles of the distance-distributions of the parameter regressors achieved within ABC for ENCA (blue) and INCA (red), as a function of the total number of used statistics, for model (13). Circles, squares and triangles correspond to the regressors for $\alpha$, $\delta$, and $\epsilon$, respectively. Latent spaces of dimension 4 and 5 yield the most accurate posterior for ENCA (Fig. 9), whereas for INCA 5 summary statistics lead to the best result (Fig. 10). The full distributions are shown in the Appendix.

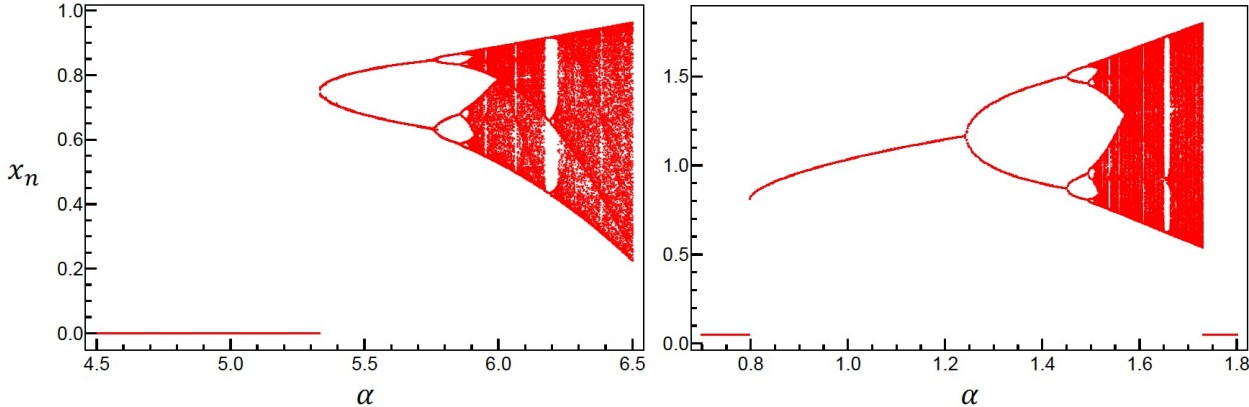

Figure 12: Amplitudes of the iterates $x_n$ as a function of the parameter $\alpha$, for the deterministic part of model (9) (left) and for model (13) (right) assuming a constant $\alpha$ and replacing the additive noise with its average value. In both cases, one observes two stable fixed points, corresponding to zero and non-zero solutions, and the cascade of period doubling bifurcations that characterizes the route to a chaotic regime.

## 6   Appendix

### 6.1   Statistical Model Priors

We conducted our experiments for eq. (9) using the following priors: $\alpha \sim \mathcal{U}(4.2, 5.8)$, $\sigma \sim \mathcal{U}(0.005, 0.025)$, and $x_0 = 0.25$. The true values used in ABC experiments were $\alpha = 5.3$ and $\sigma = 0.015$. For model (13), we used the following priors: $\alpha \sim \mathcal{U}(0.9, 1.4)$, $\delta \sim \mathcal{U}(0.05, 0.25)$, $\epsilon \sim \mathcal{U}(0.02, 0.15)$, and $x_0 = 1.0$. The true values used in ABC experiments were $\alpha = 1.11$, $\delta = 0.15$ and $\epsilon = 0.08$.

The choice of statistical model priors and true values in our experiments are based on the stable points, bifurcations, and chaotic regions of the test models (see Fig. 12). For the deterministic part of model (9), the first bifurcation occurs at $\alpha \approx 5.3$. Near this point, the iterative map has two stable fixed points - the zero solution ($\alpha \lesssim 5.3$) and a periodic solution ($\alpha \gtrsim 5.3$). Increasing $\alpha$ further, the periodic solution eventually becomes chaotic after a cascade of period-doubling bifurcations. However, for this model, we limit the prior to a range away from the chaotic regime. Adding noise to the iterative map allows for switching between the two stable solutions. We chose the true noise and the initial condition such that the variable $x_n$ randomly jumps to either one of the two stable solutions at the beginning of the time-series and then typically stays there. This is to exemplify that stochastic models can exhibit rather different types of behavior, even for fixed parameters. In the case of model (13), the parameter priors ensure that all regimes are taken into account, including the stable solutions, all bifurcations and the chaotic regime.

### 6.2   Training and evaluation details

For both INCA and ENCA, we use the same NN architecture for the encoder, which consists of only convolutional and pooling layers (see Table 1). One major difference in implementation comes from the fact that ENCA has 3 dimensional activations, (minibatch size, temporal axis, channel axis), as opposed to 4 in INCA, which has an additional dimension

| input | layer name | hyper-parameters | output shape (ENCA) | output shape (INCA) |
|---|---|---|---|---|
|  | observation | input | (bs, 200, 1) | (bs, $n$, 200, 1) |
| observation | conv1.1 | 3, 16 relu | (bs, 198, 16) | (bs, $n$, 198, 16) |
| conv1.1 | conv1.2 | 3, 16 relu | (bs, 196, 16) | (bs, $n$, 196, 16) |
| conv1.2 | maxpool | 2 | (bs, 98, 16) | (bs, $n$, 98, 16) |
| maxpool1 | conv2.1 | 3, 32 relu | (bs, 96, 32) | (bs, $n$, 96, 32) |
| conv2.1 | conv2.2 | 3, 32 relu | (bs, 94, 32) | (bs, $n$, 94, 32) |
| conv2.2 | conv3 | 3, $q$, linear | (bs, 92, $q$) | (bs, $n$, 92, $q$) |
| conv3 | globpool | N/A | (bs, $q$) | (bs, $n$, $q$) |

Table 1: Encoder architectures of the ENCA and INCA models. Global average pooling layer is denoted as globpool. Convolutional layer (conv) hyper-parameters correspond to kernel size, number of filters, and activation, respectively. Output shapes are listed for observation length of 200 steps and minibatch size is denoted as bs.

Decoder ENCA

| input | layer name | hyper-parameters | output shape |
|---|---|---|---|
| summary statistics | tile $\mathbf{s}$ | N/A | (bs, 200, $q$) |
|  | $\epsilon$ | noise | (bs, 200, $c_\epsilon$) |
| [tile $\mathbf{s}$, noise] | concatenate | N/A | (bs, 200, $c_\epsilon + q$) |
| concatenate | Bi-LSTM1 | 16, tanh | (bs, 200, 32) |
| Bi-LSTM1 | Bi-LSTM2 | 16, tanh | (bs, 200, 32) |
| Bi-LSTM2 | FC | 1, linear | (bs, 200, 1) |

Table 2: Decoder architecture of ENCA. Bidirectional (Bi)-LSTM layer details correspond to number of LSTM filters ($\times 2$ due to Bi) and activation, respectively. Fully connected (FC) layer details correspond to number of hidden units, and activation, respectively. Output shapes are listed for an initial observation length of 200 steps constructed using noise vector $\epsilon \in \mathbb{R}^{200 \times c_\epsilon}$ having $c_\epsilon$ channels. Minibatch size is denoted as bs. Note that $c_\epsilon$ is 1 and 2, for the two test models, respectively. Tile layer corresponds to broadcasting operation along a new axis. For more details, please refer to our repository.

Decoder INCA

| input | layer name | hyper-parameters | output shape |
|---|---|---|---|
|  | $\{\mathbf{s}^{(j)}\}_{j=p+1,\dots,q}$ | N/A | (bs, $n$, $q-p$) |
| $\{\mathbf{s}^{(j)}\}_{j=p+1,\dots,q}$ | FC1 | 3, leakyReLU | (bs, $n$, 3) |
| FC1 | FC2 | 10, leakyReLU | (bs, $n$, 10) |
| FC2 | FC3 | 3, leakyReLU | (bs, $n$, 3) |
| FC3 | FC4 (i.e., $w(\cdot)$) | 1, sigmoid | (bs, $n$, 1) |
| $[w(\cdot), \{\mathbf{s}^{(j)}\}_{j=1,\dots,p}]$ | Eq. (7) | N/A | (bs, $p$) |

Table 3: Decoder architecture of INCA. Fully connected (FC) layer details correspond to number of hidden units, and activation, respectively. All leakyReLU activations have their $\alpha = 0.3$. Output shapes are listed for an initial observation length of 200 steps constructed using noise vector $\epsilon \in \mathbb{R}^{200 \times c_\epsilon}$ having $c_\epsilon$ channels. Minibatch size is denoted as bs. $\{\mathbf{s}^{(j)}\}_{j=p+1,\dots,q}$ corresponds to auxiliary summary statistics. For more details, please refer to our repository.

for the $n$ replica. The decoder architectures for ENCA and INCA are shown in Tables 2 & 3. We chose a recurrent neural network architecture for the ENCA-decoder in order to estimate the input observation where we paired bare noise with summary statistics for each time step. We used a multilayer perceptron architecture to learn the mapping function $w(\cdot)$ within the INCA-decoder, which is later used as in eq. (7).

For training ENCA to model (9), we simulated, on-the-fly and per training step, a minibatch of 300 model realizations, that is, 300 triplets of parameter vectors, noise vectors and associated model outputs. Per training step of INCA, we simulated a minibatch of $n = 5$ replica of 60 model realizations each. That is, we used 300 pairs of parameters and associated model outputs per training step. For model (13) instead we set the number of model realizations per training step to 100 for ENCA, and to $5 \times 60 = 300$ for INCA. We used the Adam optimizer with an initial learning rate of $10^{-3}$ for training both architectures.

We trained the networks well beyond convergence of their loss functions. A retrospective analysis shows that the loss curves were already at the plateau at approximately $7 \cdot 10^5$ (ENCA) and $10^6$ (INCA) steps for model (9), and $5 \cdot 10^5$ steps for both ENCA and INCA for model (13). This implies that on the order of 50M to 300M observations were generated on the fly to train each network. For more expensive simulation models, we recommend to pre-compute a much more modest number of independent model realizations, and then train the networks for several epochs on this dataset until the loss function has converged.

The proposed networks ENCA and INCA have less than 15k and 6k network weight parameters, respectively, to be trained for the test models in this work. These numbers are invariant to the time-series length of observed samples. This implies that our proposed networks can be used to train on significantly longer samples without requiring more network parameters to be learned unless an architectural change is deemed necessary.

For the ABC inference we used the simulated annealing ABC algorithm [Albert et al., 2014] as implemented in the SPUX[2] framework. Each inference took about $75 \cdot 10^4$ model realisations, for both models. This large number of model realisations ensures that the ABC-inference process practically converges, with a remaining acceptance rate of typically well below 1%.

---

[2]https://spux.readthedocs.io

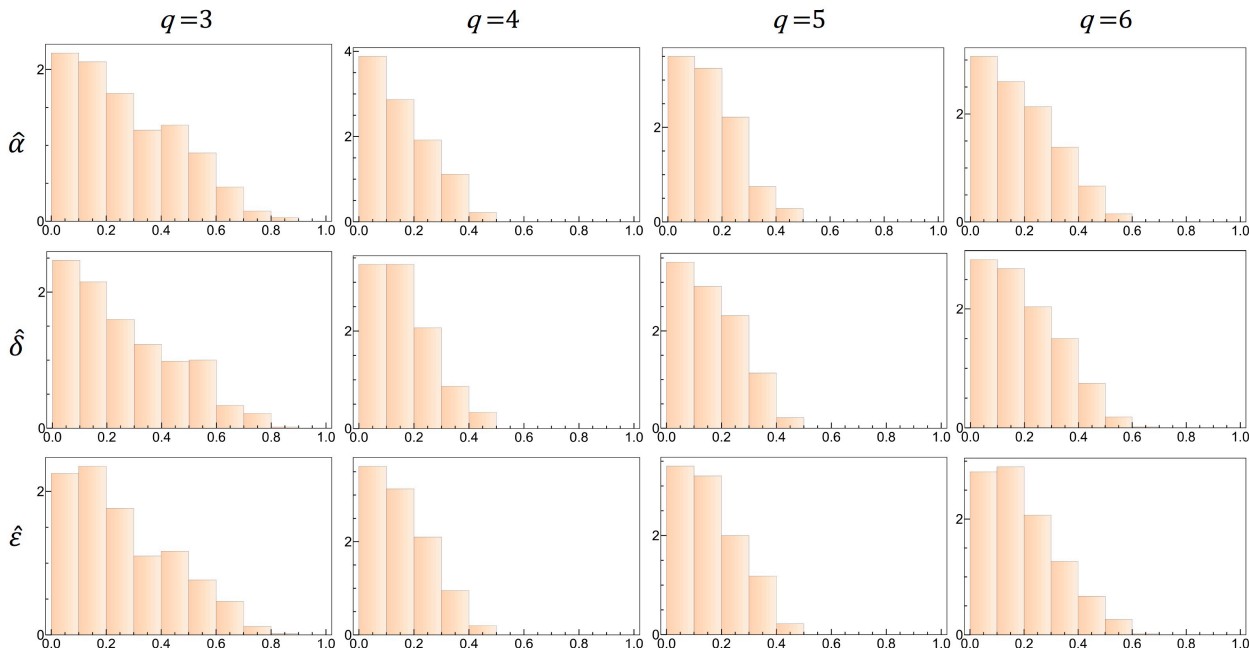

Figure 13: Histograms of the final accepted distances between observed and simulated summary statistics achieved with ABC, for model (13). Only the first three summary statistics, i.e., the parameter regressors, are plotted (rows). The columns correspond to different architectures of ENCA, that is, latent space dimensions of 3, 4, 5, or 6, as specified by the column labels.

The training of the Autoencoders and the ABC inference were run on the internal cluster of the Zurich University of Applied Science (ZHAW, Switzerland), using for each of them a full node equipped with two 16-core 2.6-3.7 GHz processors (Xeon-Gold 6142) and 196 GB of memory. On this infrastructure, training the networks takes about 4 days for both ENCA and INCA for model (9), and 3 (ENCA) and 7 (INCA) days for model (13). The ABC inference takes about 2 hours. Both training and inference are automatically parallelized across the available CPU cores by tensorflow (version 2.2) and SPUX, respectively.

### 6.3   Additional results

Figs. 13 and 14 show the full histograms of the ABC-distances, from which the quantiles in Fig. 11 in the main text were derived.

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

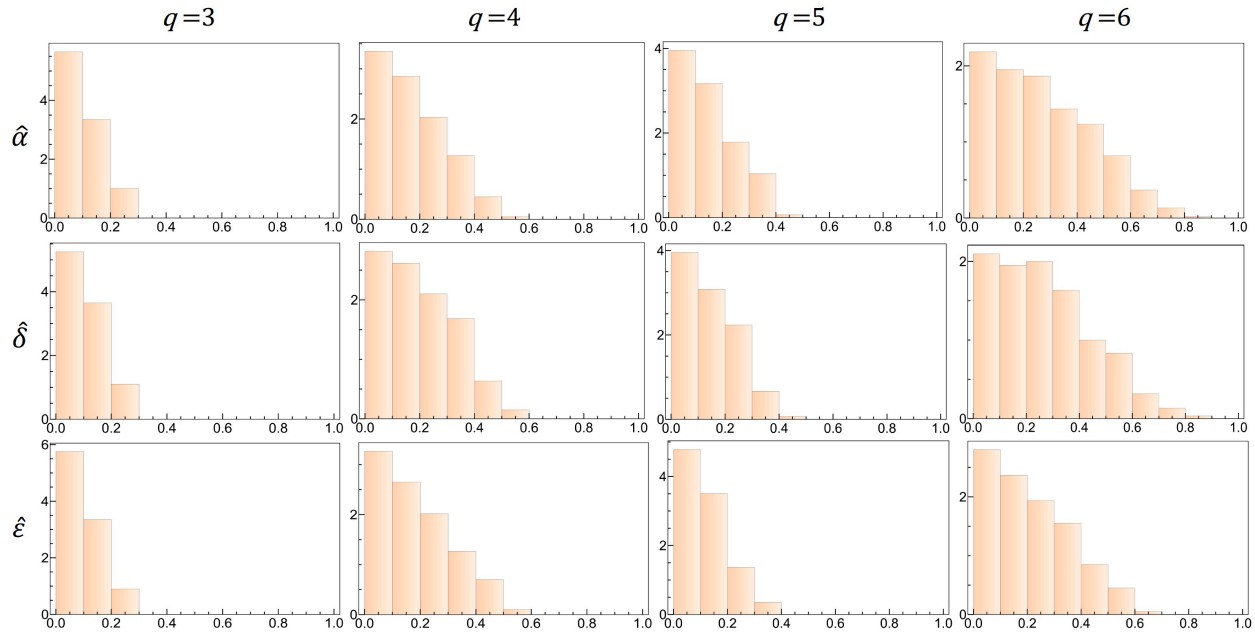

Figure 14: Distance distributions as in Fig. 13, for different architectures of INCA.

[Mandelbrot, 1962] Mandelbrot, B. (1962). The role of sufficiency and of estimation in thermodynamics. *The Annals of Mathematical Statistics*, pages 1021–1038.

[Marin et al., 2012] Marin, J., Pudlo, P., Robert, C., and Ryder, R. (2012). Approximate Bayesian computational methods. *Statistics and Computing*, 22(6, SI):1167–1180.

[Papamakarios and Murray, 2016] Papamakarios, G. and Murray, I. (2016). Fast $\varepsilon$-free inference of simulation models with Bayesian conditional density estimation. In *NIPS*.

[Papamakarios et al., 2019] Papamakarios, G., Sterratt, D., and Murray, I. (2019). Sequential neural likelihood: Fast likelihood-free inference with autoregressive flows. In *The 22nd International Conference on Artificial Intelligence and Statistics*, pages 837–848. PMLR.

[Tavaré et al., 1997] Tavaré, S., Balding, D., Griffiths, R., and Donnelly, P. (1997). Inferring Coalescence Times From DNA Sequence Data. *Genetics*, 145:505–518.

[Wetzel, 2017] Wetzel, S. J. (2017). Unsupervised learning of phase transitions: From principal component analysis to variational autoencoders. *Physical Review E*, 96(2):022140.

[Wiqvist et al., 2019] Wiqvist, S., Mattei, P.-A., Picchini, U., and Frellsen, J. (2019). Partially exchangeable networks and architectures for learning summary statistics in approximate Bayesian computation. In *International Conference on Machine Learning*, pages 6798–6807. PMLR.