# Peer review of "Learning Summary Statistics for Bayesian Inference with Autoencoders"

_SciPost Physics Core_

## Round 1 · Referee Report · Anonymous (Referee 1) · 2022-2-24

Strengths

1 - The paper is clearly written with the "2- summary statistics" section providing intuitions for the design choices made by the authors in their method.
2 - It presents two new methods and exemplify their effectiveness.

Weaknesses

1 - Experiments could be further developed, in particular the authors do not compare, nor clarify why the comparison is not relevant, with other methods (such as the Cvitkovic et al 2019).
2 - Some information about experiments appear to be missing (see report).
3 - A discussion about scalability/computational cost would also have been a good addition (cf several days of training for the presented experiments).

Report

The present work introduces a strategy to learn summary statistics to infer the parameters of a forward model given available data using ABC. The authors motivate their approach by identifying desirable properties of sufficient statistics: asymptotic sufficiency and asymptotic minimality formulated in terms of information theoretic quantities. More precisely the authors propose two strategies ENCA and INCA.

For ENCA, one assumes that the forward model can be written as a deterministic function of the parameters and a noise variable. The sufficient statistics are extracted from the latent space of an autoencoder of the data for which the decoder is also fed the noise variable. The sufficiency is encouraged through a data reconstruction term in the training loss. The minimality is encouraged by a parameter regression term forcing a subset of the coordinates of the summary statistic to fit the parameters.

For INCA, instead of feeding directly the noise variable to the model, implicit information is passed by considering batches of simulation outputs produced with the same value of the parameter. This time the “autoencoder” is on the parameter space and includes the forward model in the encoder. Again, the summary statistics is extracted as the latent space and a part of its coordinates is encouraged to act as a parameter regressor. The decoder then takes the form of a weighted sum over the parameter estimate on the batch of inputs with coefficients produced as non-linear functions of the summary statistics.

The paper finally presents numerical results on two stochastic iterative maps. For one model, they show agreements between true parameter and parameter fits in the trained summary statistics similar to MLEs and more concentrated ABC samples compared to the use of parameter regressors only. For the other model, they show good agreement between ABC samples and exact posterior samples, as well as better reconstruction capacity of the decoder from learned summary statistics bigger than the parameter regressor.

The paper is well motivated and to the best of my knowledge original. Presented experiments are sound. Overall it is clearly written although a few clarifications on the experimental section would be helpful (see below). Upon revision to clarify the suggested points, I think the paper is suitable for publication.

Clarifications:
I assume that the training data are generated by randomly sampling values of parameters and using the forward model to obtain corresponding data, creating (parameter, data) pairs which can be fed in the proposed training objective. I would recommend a sentence to explicitly state this procedure (unless it is already included and I somehow missed it). Along the same line, to the best of my understanding the authors have provided training time but did not state the number of training datapoints necessary, which I think is an interesting order of magnitude to provide. Similarly, have the authors provided the dimensionality of the data they are using (N)?

I also found a reference to "our repository" in figures' caption, but not the link to the said repository - unless it is the same as for the ABC framework (https://spux.readthedocs.io ), please clarify where to find the codes.

Requested changes

Minor changes: 1- I assume that the training data are generated by randomly sampling values of parameters and using the forward model to obtain corresponding data, creating (parameter, data) pairs which can be fed in the proposed training objective. I would recommend a sentence to explicitly state this procedure (unless it is already included and I somehow missed it). 2- Along the same line, to the best of my understanding the authors have provided training time but did not state the number of training datapoints necessary, which I think is an interesting order of magnitude to provide. 3- Similarly, have the authors provided the dimensionality of the data they are using (N)? 4 - I also found a reference to a repo, but not the link - unless it is the same as for the ABC framework (1 https://spux.readthedocs.io ), please clarify where to find the codes.

  • validity: high
  • significance: -
  • originality: -
  • clarity: high
  • formatting: -
  • grammar: perfect

Author:  Carlo Albert  on 2022-05-24  [id 2515]

(in reply to Report 1 on 2022-02-24)

Weaknesses 1 - Experiments could be further developed, in particular the authors do not compare, nor clarify why the comparison is not relevant, with other methods (such as the Cvitkovic et al 2019).

We agree that further experiments and comparisons are warranted, and we will address them in future publications. The main goal of this publication is to present a new method for disentangling low-dimensional relevant features from high-dimensional irrelevant (noise) features. As our method is also applicable to observed data, for which no parameters are available, the potential range of applications goes beyond Bayesian inference. This is not the case for the other methods we are referring to. In order to stress this point, we have added the following paragraph to the introduction:

“What distinguishes our approach from other information-theoretic machine learning algorithms such as the ones recently presented in \cite{cvitkovic_2019_MinSuffStatML} and \cite{chen2020MLsufficientStats}, is the possibility to use {\em explicit} noise information. This makes our approach also applicable in situations where only noise-, but no parameter-information is available, as could be the case in observed rather than simulated data. As an example, we might want to use it to remove rain-features (rain playing the role of the noise) from hydrological runoff time-series in order to distill runoff-features that stem from the catchments themselves. We will examine this application in future publications.”

2 - Some information about experiments appears to be missing (see report).

Please refer to our answers below.

3 - A discussion about scalability/computational cost would also have been a good addition (cf several days of training for the presented experiments).

We agree that instead of only mentioning the resources we used for the experiments, it would be valuable to comment on the required computational costs and scalability concerns. Training of both ENCA and INCA for both models is computationally cheap with regards to the required hardware. ENCA and INCA have less than 15k and 6k network weight parameters to be trained, respectively. Given that training data is generated on-the-fly, both neural networks can be trained on modest hardware such as publicly available resources (e.g., https://renkulab.io/projects/bistom/enca-inca/sessions/new?autostart=1). This leaves the training time of 3-7 days as the only computational cost, which can be further shortened if more computational nodes in parallel are available. Regarding scalability, there are no concerns. We can consider the following scenario where a new model may have significantly longer time-series as observations as opposed to N=200. For both ENCA and INCA, the encoders apply a fixed number of convolutions and max-pooling, followed by a global average pooling (c.f. Appendix, Table 1). So the number of parameters in the encoders are invariant to the length of the time-series. The decoder of ENCA utilizes LSTMs (c.f. Appendix, Table 2). This implies that while the number of learned parameters is invariant to the length of the timeseries, the duration to reconstruct the time-series from latent embedding would linearly increase based on the length of the time-series. However, as long as the observed samples are not in the order of million time steps, it would not be creating a significant concern for scalability. There are less than 100 parameters to be trained within the decoder of INCA (c.f. Appendix, Table 3), which consists of an MLP acting on the auxiliary summary statistics to learn the w(.) function. Accordingly, the number of parameters in the decoder of INCA is invariant to the length of the time-series. We now stress these points in the revised manuscript in Section 6.2:

“We trained the networks well beyond convergence of their loss functions. A retrospective analysis shows that the loss curves were already at the plateau at approximately $7\cdot 10^5$ (ENCA) and $10^6$ (INCA) steps for model (9), and $5 \cdot 10^5$ steps for both ENCA and INCA for model (13). This implies that on the order of 50M to 300M observations were generated on the fly to train each network. For more expensive simulation models, we recommend to pre-compute a much more modest number of independent model realizations, and then train the networks for several epochs on this dataset until the loss function has converged. The proposed networks ENCA and INCA have less than 15k and 6k network weight parameters, respectively, to be trained for the test models in this work. These numbers are invariant to the time-series length of observed samples. This implies that our proposed networks can be used to train on significantly longer samples without requiring more network parameters to be learned unless an architectural change is deemed necessary.”

Requested changes

Minor changes:

1- I assume that the training data are generated by randomly sampling values of parameters and using the forward model to obtain corresponding data, creating (parameter, data) pairs which can be fed in the proposed training objective. I would recommend a sentence to explicitly state this procedure (unless it is already included and I somehow missed it).

That is correct. In the revised version, we have described the training procedure more explicitly in the Appendix. We modified Sec 6.2 in the Appendix to explicitly state what a minibatch consists of for ENCA and INCA.

“For training ENCA to model (9), we simulated, on-the-fly and per training step, a minibatch of 300 model realizations, that is, 300 triplets of parameter vectors, noise vectors and associated model outputs. Per training step of INCA, we simulated a minibatch of $n=5$ replica of 60 model realizations each. That is, we used 300 pairs of parameters and associated model outputs per training step. For model (13) instead we set the number of model realizations per training step to 100 for ENCA, and to $5\times 60=300$ for INCA.”

2- Along the same line, to the best of my understanding the authors have provided training time but did not state the number of training datapoints necessary, which I think is an interesting order of magnitude to provide.

The number of data points we have used for training is indeed somewhat hidden in the Appendix. We have written this more clearly in the revised manuscript. See our citations in the answers above.

3- Similarly, have the authors provided the dimensionality of the data they are using (N)?

The dimension of the data N=200 was also a bit hidden in the Appendix. We have added this information to the main text.

4 - I also found a reference to a repo, but not the link - unless it is the same as for the ABC framework (1 https://spux.readthedocs.io ), please clarify where to find the codes.

That is correct. We initially added a link to our repository through paperswithcode.com on arXiv, however, this link has disappeared due to a technical problem. We have fixed this in the meantime. Nevertheless, we have added a reference to our repository at the end of the Introduction.

---

## Round 1 · Referee Report · Anonymous (Referee 2) · 2022-3-25

Strengths

1- The paper seems technically correct 2- (I guess) that the method is useful when the likelihood estimators are difficult to obtain

Weaknesses

1- The context of the method is correctly presented, but I feel like there is a lack of motivation. In particular, the number of refences is quite low, ~10. Still, I'm not an expert in this field but it still feels that some information and references on where/why it is useful and/or practical is missing. 2- In relation with the previous remark, the chosen example are not motivated clearly to me. 3- A part of the discussion is about the need to add "auxiliary" summary statistics, however it is not clear how to do it in general ?, in the first example a choice is made based upon prior knowledge of the problem. In the second one, it is not even discussed. 4- the presentation of the results- how are made the experiments- is not clear to me.

Report

This article presents a method to reconstruct the parameters of stochastic processes without the need of performing costly Monte Carlo simulation to approximate the posterior distribution. The methods rely on the use of AutoEncoder where the value of the parameters is reconstructed. Additionally, it works by potentially adding more parameters to reconstruct than the one present in the model to deal with stochastic process that can need many sufficient statistics to be characterized.

The perspective and motivation of the presented work is hard to grasp for me, but it is true that I'm not an expert in the field. From the point of view of the thematic, I'm not sure why this work has been sent to a physics journal, while most of the discussions are not related to any physical process or even to statistical physics. Only a small paragraph making a quick comment on the equivalence between the micro-canonical and the canonical ensemble seems relevant here.

Concerning the work presented, again I'm not an expert, but I found that several points are hard to grasp or misleading for the reader. First, the experimental protocol is only barely discussed in the appendix. The MLE method is not described, and again it is not clear to me how are obtained the presented results: is there only one AE inferring the model's parameters ? of several ? How the training is done ? presenting minibatches for each value of alpha ? etc.
Two examples are illustrated here. Why they have been chosen remain mysterious to me. Does there exist more practical examples on which the method could be applied ?
Also, for the ENCA model, how are defined the auxiliary parameters ?

My overall opinion (apart from the thematic matching) is that this paper should first clarify the definition of the model (particularly the second one), and provide more relevant experiments to be published.

Requested changes

1- I'm confused with the model of eq 10. As far as I understand, it is a nonlinear mapping showing one or two solutions as discussed in the article. However, I do not know to what transition to chaos the authors refer to. Can it be made more explicit ? (even the appendix remained somehow mysterious on this point).

2- The second model is quite different from the first one, and it is not clear what motivated the changes. For instance, why not doing the reconstruction \hat{x} of the input ? Why using eq 8 for the "decoder" and not a classical neural network ?

3- on fig 5., why not putting an inset on the interesing region ?

4- It is not clear to me if the results of MLE is using some kind of neural network giving only the eq. 11 and 12 as parameters to reconstruct, or more simply the estimation given directly by eq 11 and 12. It can be appropriate to write it explicitly somewhere.

5- I'm confused with fig 5, what is the metropolis-generated posterior ? and how it is computed ? It seems to give better results than the method presented here. It is ok if it is to compare to other methods but then you should discuss why using the one presented in this work has some advantage.

6- In the first experiment, no result is given where q is higher than p. Is it because it doesn't improve at all ?

7- About fig 7: why the reconstruction is not illustrated in the first example ? It is not mentioned that the reconstruction is done on a set of data that have not been used during the training. Could you be more precise on that.

  • validity: good
  • significance: ok
  • originality: good
  • clarity: low
  • formatting: good
  • grammar: good

Author:  Carlo Albert  on 2022-05-24  [id 2514]

(in reply to Report 2 on 2022-03-25)
Category:
answer to question

Weaknesses 1- The context of the method is correctly presented, but I feel like there is a lack of motivation. In particular, the number of refences is quite low, ~10. Still, I'm not an expert in this field but it still feels that some information and references on where/why it is useful and/or practical is missing.

Bayesian inference with stochastic models has a large number of practical applications. Unfortunately, the Bayesian posterior typically depends on very high-dimensional integrals. Therefore, finding good summary statistics is critical for all methods that attempt to circumvent direct numerical evaluation of these integrals. Machine learning offers new and exciting methods of learning such statistics, and our contribution goes in this direction. We do not want to give an exhaustive overview over the vast literature on this problem, but rather cite some key papers in which the reader can find more relevant references. In the revised manuscript, we have added another key reference ([Greenberg et al., 2019]). Because one of our methods, ENCA, does not necessarily require access to the model parameters, it can potentially be applied in a much broader context of feature learning. We have added this outlook to the Introduction:

“What distinguishes our approach from other information-theoretic machine learning algorithms such as the ones recently presented in \cite{cvitkovic_2019_MinSuffStatML} and \cite{chen2020MLsufficientStats}, is the possibility to use {\em explicit} noise information. This makes our approach also applicable in situations where only noise-, but no parameter-information is available, as could be the case in observed rather than simulated data. As an example, we might want to use it to remove rain-features (rain playing the role of the noise) from hydrological runoff time-series in order to distill runoff-features that stem from the catchments themselves. We will examine this application in future publications.”

2- In relation with the previous remark, the chosen example are not motivated clearly to me.

The motivation of the first example (eq. (9) in the revised manuscript) is to use a simple stochastic model that can exhibit very different behavior, even for fixed parameters (akin to phases in statistical mechanics). In such cases, it is typically necessary to use more summary statistics than parameters in order to distinguish these different types of behavior. The motivation behind the second model (eq. (13)) is to use a simple example for the common situation where we have more noise degrees of freedom (epsilon and alpha time series) than outputs (x time series). Integrating out the excess degrees of noise typically takes us outside of the exponential family where there is no bounded set of sufficient summary statistics any more. Our results show that, even in this situation, a satisfactory approximation of the posterior is achieved with a small number of statistics and that our proposed methods can find them.

3- A part of the discussion is about the need to add "auxiliary" summary statistics, however it is not clear how to do it in general ?, in the first example a choice is made based upon prior knowledge of the problem. In the second one, it is not even discussed.

We have chosen the first example because (i) it has a finite number of sufficient statistics and (ii) this number is larger than the number of parameters. In this example, we have access to analytical equations for such a set of statistics, and we can compare the summary statistics learned by our Autoencoder with them. In the second example, we know that no bounded set of sufficient statistics exists and therefore q has to be set to N to reach strict sufficiency. However, our results show that our Autoencoder is capable of packing enough information into a small number of statistics to reach a satisfactory approximation of the true posterior (which we also know in our example). Notice that all we have to do is to decide on the number of auxiliary statistics (dimension of the bottleneck). The architecture of our Autoencoder is such that the decoder creates the incentive for the encoder to encode as much parameter-related information as possible into these statistics. Please notice that in the last paragraph of chapter 4 we did discuss the question of the optimal number of summary statistics, and we offer an empirical criterion based on the distances between simulated and observed summary statistics achieved within ABC. However, finding the optimal number of statistics remains a difficult question that warrants further research.

4- the presentation of the results- how are made the experiments- is not clear to me.

In the Appendix 6.2 of the revised version, we are now much more explicit about the implementational details:

“For training ENCA to model (9), we simulated, on-the-fly and per training step, a minibatch of 300 model realizations, that is, 300 triplets of parameter vectors, noise vectors and associated model outputs. Per training step of INCA, we simulated a minibatch of $n=5$ replica of 60 model realizations each. That is, we used 300 pairs of parameters and associated model outputs per training step. For model (13) instead we set the number of model realizations per training step to 100 for ENCA, and to $5\times 60=300$ for INCA. We used the Adam optimizer with an initial learning rate of $10^{-3}$ for training both architectures.”

Report

This article presents a method to reconstruct the parameters of stochastic processes without the need of performing costly Monte Carlo simulation to approximate the posterior distribution. The methods rely on the use of AutoEncoder where the value of the parameters is reconstructed. Additionally, it works by potentially adding more parameters to reconstruct than the one present in the model to deal with stochastic process that can need many sufficient statistics to be characterized. The perspective and motivation of the presented work is hard to grasp for me, but it is true that I'm not an expert in the field. From the point of view of the thematic, I'm not sure why this work has been sent to a physics journal, while most of the discussions are not related to any physical process or even to statistical physics. Only a small paragraph making a quick comment on the equivalence between the micro-canonical and the canonical ensemble seems relevant here.

We found the equivalence between thermodynamic state variables in statistical mechanics and minimally sufficient statistics in the broader context of stochastic models very appealing, which is why we decided to send our manuscript to a physics journal. In the revised version, we have extended the analogy between statistical mechanics and the problem of finding summary statistics for generic stochastic models further:

“The asymptotic sufficiency and concentration properties allow us to draw an analogy between summary statistics and {\em thermodynamic state variables}, as already pointed out by \cite{mandelbrot_1962_sufficiencyThermodynamics}. In statistical mechanics, the concentration property leads to the {\em equivalence of ensembles} in the thermodynamic limit. Extending this analogy a bit further, we define the {\em free energy} \begin{equation} F_{\boldsymbol{\theta}}({\bf s}):=-\ln \int f({\bf x}|\boldsymbol{\theta}) d\Omega_{{\bf s}}({\bf x}) \,, \end{equation} where $\Omega_{{\bf s}}({\bf x})$ is the surface measure on the shell ${\bf s}({\bf x})={\bf s}$. For members of the exponential family, the free energy splits into an energy and an entropy term as \begin{equation} F_{\boldsymbol{\theta}}({\bf s}) = U_{\boldsymbol{\theta}}({\bf s})-S({\bf s}) \,. \end{equation} If the data ${\bf x}$ is comprised of a large number of {\em independent} sample points, the free energy will be dominated by the energy term. Hence, the summary statistics will concentrate around a $p$-dimensional submanifold of minimal energy configurations. Since minimum energy is synonymous with maximum likelihood, we can parametrize this manifold with {\em maximum likelihood estimators} (MLE). Hence, we can encode nearly all information relevant for constraining the parameters with $q=p$ summary statistics. This is no longer the case if ${\bf x}$ is {\em correlated}, in which case the entropy can substantially alter the free energy landscape. If (2) is satisfied, the summary statistics will eventually concentrate locally around a $p$-dimensional submanifold. However, due to correlations, certain features of the output might de-correlate very slowly and lead to broad valleys or multiple modes in the free energy landscape. Multiple modes can even persist in the limit as $N\rightarrow\infty$, in which case the model exhibits different {\em phases}. For such models, ${\bf S}$ does not concentrate around a single $p$-dimensional submanifold, and we might need $q>p$ summary statistics, for a good posterior approximation. “

Concerning the work presented, again I'm not an expert, but I found that several points are hard to grasp or misleading for the reader. First, the experimental protocol is only barely discussed in the appendix. The MLE method is not described, and again it is not clear to me how are obtained the presented results: is there only one AE inferring the model's parameters ? of several ? How the training is done ? presenting minibatches for each value of alpha ? etc.

Please notice that MLE does not refer to a method, but simply refers to eqs. (10) and (11). That is, the MLE results refer to ABC-results achieved with (10) and (11) as the only two summary statistics. There are two different autoencoders, ENCA and INCA. Both are able to learn near-sufficient summary statistics, for both models (9) and (13). Regarding the training of the Autoencoders, we are now much more explicit about it in the Appendix (see the citation above).

Two examples are illustrated here. Why they have been chosen remain mysterious to me. Does there exist more practical examples on which the method could be applied ?

Please refer to our comments above. The examples were chosen because, for both models, (i) we have access to the exact posteriors which allows us to compare our approximate ABC results and (ii) we need strictly more summary statistics than parameters. For the first example we know a sufficient set of summary statistics and can compare our learned statistics against them, for the second example there is no bounded set of summary statistics, and yet we show that a small number of learned statistics does a good job in approximating the posterior. More practical applications will be addressed in future publications. This paper is meant to be a proof-of-concept for the method.

Also, for the ENCA model, how are defined the auxiliary parameters ?

The auxiliary summary statistics are not defined a priori. We only have to decide on the number of auxiliary statistics. Then, the AE does the rest. The idea is that the decoder creates an incentive for the encoder to use the auxiliary statistics to encode ideally all parameter-related information in the output of the stochastic model.

My overall opinion (apart from the thematic matching) is that this paper should first clarify the definition of the model (particularly the second one), and provide more relevant experiments to be published.

We hope that we could make things clearer with our answers and modifications of the manuscript. More experiments are in the making, but we hope that the experiments presented in this draft may suffice as a proof of concept.

Requested changes

1- I'm confused with the model of eq 10. As far as I understand, it is a nonlinear mapping showing one or two solutions as discussed in the article. However, I do not know to what transition to chaos the authors refer to. Can it be made more explicit ? (even the appendix remained somehow mysterious on this point).

For small values of alpha, the deterministic part of our model only has the trivial zero solution. For larger values of alpha, a second stable solution emerges. Depending on the value of alpha, this second stable solution can be either a fixed point, periodic, or chaotic (see the Fig. referred to in 6.1). For our first experiment (eq (10)), however, we constrain the prior to a range of alphas away from the chaotic regime. However, the transition to chaos is not so important here. What is important is that model realizations can go to either of two attractors, even for fixed parameters. We have added a new figure (Fig. 5) showing these two types of behavior. For the sake of inferring the model parameters, it is important to know which attractor the observed data belongs to, and this is exactly where the third summary statistic comes in.

2- The second model is quite different from the first one, and it is not clear what motivated the changes. For instance, why not doing the reconstruction \hat{x} of the input ? Why using eq 8 for the "decoder" and not a classical neural network ?

We suppose you are referring to INCA. INCA is meant for applications where we do not have easy access to the noise that was used to generate the model outputs (x). Without the explicit noise information, reconstructions \hat x would be very poor, which is why we reconstruct the true parameters instead. Since the first p latent variables are already, by design, parameter regressors, we can use equation (8) for this reconstruction. The non-trivial weighting functions are represented by neural network models. In the revised version, we have added some more clarification:

“[...] Since we do not have access to the bare noise, we do not attempt to reconstruct the model output ${\bf x}$, as such a reconstruction would generally be very poor. Instead, the decoder attempts to aggregate the information in the replica of summary statistics and reconstruct the true parameters, $\hat{\boldsymbol\theta}(\lbrace {\bf s}^{(j)}\rbrace)$.”

3- on fig 5., why not putting an inset on the interesing region ?

We have tried to zoom into the interesting region, but figured that it does not provide additional visual information.

4- It is not clear to me if the results of MLE is using some kind of neural network giving only the eq. 11 and 12 as parameters to reconstruct, or more simply the estimation given directly by eq 11 and 12. It can be appropriate to write it explicitly somewhere.

It is the latter. We directly use equations (11) and (12) for the leftmost panels in Fig. 3. We have clarified this in the caption of Fig. 3:

“MLEs (using eqs. (11 and (12)) for the parameters of model (10) (left panels),[...]”

5- I'm confused with fig 5, what is the metropolis-generated posterior ? and how it is computed ? It seems to give better results than the method presented here. It is ok if it is to compare to other methods but then you should discuss why using the one presented in this work has some advantage.

For our examples, we have access to the analytical likelihood function. Therefore, we can also sample the posterior directly, e.g. with the Metropolis algorithm. The result is the ground truth we compare our ABC results against. Of course, for most interesting stochastic models the likelihood is prohibitively expensive and we do not have access to the ground truth. However, for our proof of concept, we deemed it appropriate to use case studies where we do have the ground truth.

6- In the first experiment, no result is given where q is higher than p. Is it because it doesn't improve at all ?

This might be a misunderstanding. In the first experiment, p=2 (alpha and sigma), but the minimal number of sufficient statistics is q=3. We have not gone higher than q=3 as the result for q=3 is already very good. However, in the revised version, we added the results for q=2. As expected they are worse than the q=3 results, however better than the MLE results.

7- About fig 7: why the reconstruction is not illustrated in the first example ? It is not mentioned that the reconstruction is done on a set of data that have not been used during the training. Could you be more precise on that.

That is a good point. We have now added the reconstruction plots for the first example as well. We are now also mentioning in the captions of the figures that the reconstructions we show are done on data that has not been used for training before.

---

## Round 2 · Referee Report · Anonymous · 2022-6-7

Strengths

1. The authors improved their manuscript given the referee's remark. I feel that the article is clearer now.
2. They add some other results confirming their previous results.

Report

The authors have answered positively to the referee's comment. On my side, as not expert on the field, I see no particular reason not to accept the manuscript.

---

## Round 2 · Referee Report · Anonymous · 2022-6-11

Report

The others answered to the question I raised and in particular clarify points that were originally missing in the manuscript. As such, I recommend publication.

---

## Round 2 · Author Response

Based on the reviewer comments, we have substantially reworked our manuscript. We have replied to all the issues raised by the two reviewers using the reply option on the submission page.

---

## Round 2 · List of Changes

1) We have modified our definition of the concentration property. In the previous version, we had used a definition that worked for discretized variables only.

2) We have removed a minor inconsistency between the manuscript and the code. The nonlinear function we are using in Eq.9 is f(x) = x(1-x), not f(x) = x^2exp(-x) as previously written in the manuscript.

3) We added a distinguishing feature of our approach to the Introduction:

“What distinguishes our approach from other information-theoretic machine learning algorithms such as the ones recently presented in \cite{cvitkovic_2019_MinSuffStatML} and \cite{chen2020MLsufficientStats}, is the possibility to use {\em explicit} noise information. This makes our approach also applicable in situations where only noise-, but no parameter-information is available, as could be the case in observed rather than simulated data. As an example, we might want to use it to remove rain-features (rain playing the role of the noise) from hydrological runoff time-series in order to distill runoff-features that stem from the catchments themselves. We will examine this application in future publications.”

4) At the end of Section 2, we further extended the analogy between summary statistics learning and thermodynamics. In particular, we stress the importance of considering the entropy of summary statistics, when doing inference with high-dimensional correlated data.

5) We added a new figure (Fig. 5), showing the reconstruction capabilities of the decoder for the first model.

6) We added new inference results, for the case when only 2 summary statistics are learned, for the first model: Figs. 5 and 6 (left panel).

7) In Appendix 6.2, we are much more explicit about how the Autoencoders are trained. We also added a short discussion on the scalability of our methods.

---

## Editorial Decision

publication_decision_taken:_accept